# Thirty years of declining stunting in Tanzania: Trends and ongoing challenges

Ester Elisaria[1], Bet Caeyers[2,3], Esther Nkuba[4], Laura van der Erve[5], August Kuwawenaruwa[1]*

1 Ifakara Health Institute, Dar es Salaam, Tanzania, 2 Chr. Michelsen Institute (CMI), Bergen, Norway, 3 Centre for Experimental Research on Fairness, Inequality and Rationality (FAIR) & Norwegian School of Economics, 4 Tanzania Food and Nutrition Center (TFNC), Dar-es-Salaam, Tanzania, 5 Oxford Policy Management, Oxford, United Kingdom,

* ajoachim@ihi.or.tz

## Abstract

### Background

Tanzania has made considerable gains in children's nutrition between 1991/92 and 2022. The country's stunting rate has decreased from 50% in 1992 to 30% in 2022. However, stunting varies greatly among regions. The purpose of this study was to examine regional trends in stunting rates and associated characteristics related to the risk of being stunted among children under the age of five in Tanzania.

### Methods

Descriptive statistics were used to summarize the data on stunting, from the Tanzania Demographic and Health Survey (TDHS) data. A total of 42,408 under-five children from repeated TDHS cross-sectional studies conducted in 1991–1992 (n = 6,585), 1996 (n = 5,438), 1999 (n = 2,555), 2004–2005 (n = 7,230), 2009–2010 (n = 6,792), 2015–2016 (n = 9,001), and 2022 (n = 4,807) were analysed to examine trends in stunting and its associated characteristics in the country. Frequencies and percentages were calculated and presented in tables and graphs as cross-sectional data points. A multivariable logistic regression model was estimated to identify factors associated with stunting in 2022. All analyses have been weighted using the weighting generated by the TDHS. STATA version 15 was used for data management and analysis.

### Results

Over the previous three decades, stunting rates in boys under five have consistently been 4-7 percentage points (ppt) higher than those in girls of the same age. The prevalence of stunted children was greater in households with the lowest socioeconomic status (SES) (38.6%) compared to those with the highest SES (12.6%). Similar to the trend analysis, a multivariable regression analysis found that residency in the Southern

**Data availability statement:** The dataset(s) supporting the conclusions of this article is owned by the Demographic and Health Surveys (DHS) Program, ICF, 530 Gaither Road, Suite 500, Rockville, MD 20850, USA and available upon request. To request and access the required data, visit the official webpage https://dhsprogram.com/, register, and await notification along with an official authorization letter from the DHS Program team. Further information may be requested via email from Bridgette Wellington (Data Archivist, DHS Program; Bridgette.Wellington@icf.com; & cc: archive@dhsprogram.com). The datasets used and/or analysed during the current study available from the corresponding author on reasonable request.

**Funding:** This project was funded with UK International Development from the UK government and New Zealand's Ministry of Foreign Affairs and Trade (MFAT) contract PO number 10087; however, the views expressed do not necessarily reflect the UK or New Zealand's government official policies The funders had no role in study design, data collection and analysis, decision to publish, or preparation of the manuscript.

**Competing interests:** The authors have declared that no competing interests exist.

Highlands (AOR = 2.368; 95% CI: 1.746-3.212, p < 0.001), male children (AOR = 1.583 [95% CI: 1.349 - 1.858], p < 0.001), low birth weight (AOR = 3.639 [95% CI: 2.279 - 5.810], p < 0.001), maternal exposure to alcohol (AOR = 1.440 [95% CI: 1.057 − 1.963], p < 0.05), and unimproved sanitation facilities (AOR = 1.345 [95% CI: 1.055 − 1.714], p < 0.05) were significantly associated with a higher risk of stunting. In contrast, a birth interval of 24 to 47 months (AOR = 0.762 [95% CI: 0.598 - 0.969], p < 0.001), a high level of maternal education (AOR = 0.715 [95% CI: 0.530 − 0.963], p < 0.05), and high socioeconomic status (AOR = 0.268 [95% CI: 0.178 − 0.403], p < 0.001) were significantly associated with a lower risk of stunting.

## Conclusion

Although child stunting has decreased in the country, it remains a major challenge in Tanzania, driven by factors such as residing in the Southern Highlands, child and maternal issues, and household economic and environmental factors. To combat stunting and eliminate structural obstacles, including the empowerment of marginalised groups, a multisectoral strategy is required. Furthermore, current regulations and standards place more emphasis on educating mothers about diet and health than on empowering them economically.

## Introduction

In 2022, an estimated 148.1 million children under the age of five globally were classified as stunted [1]. The prevalence of stunting was notably higher among children residing in low- and lower-middle-income countries, which accounted for 89% of the global burden of stunting in 2020. Additionally, these children were more likely to come from rural areas and have mothers who had not received an education. Alarmingly, nearly one-third of countries in Northern Africa, Oceania, and the Caribbean are experiencing an increase in the number of stunted children [2]. This trend underscores a concerning lack of progress towards the goal of halving the prevalence of stunting by the year 2030. Central Africa is the most severely affected subregion, with a prevalence rate of 37.4% [3]. In the case of Tanzania, the Demographic and Health Surveys (DHS) data collected from 1991/92 to 2022 indicates the country had a decrease in stunting from 50% in 1991/92 to 30% in 2022 [2], with a significant disparity between rural and urban areas [4]. The World Bank and UNICEF agree that heightened efforts are needed to reach the global goal of reducing the number of children suffering from stunting to 89 million by 2030 [1].

Earlier research indicated that the disparity in stunting prevalence between rural and urban areas is linked to several factors, including access to public services, education, and economic resources [4–6]. The findings further demonstrated that understanding the underlying causes of stunting in different contexts can facilitate the development of tailored interventions for both urban and rural settings [4]. Stunting occurs due to inadequate nutrition during pre-conception, pregnancy, and early

childhood [7–10]. Children affected by stunting may never reach their full potential height and may not fully develop their cognitive abilities. These children start life at a significant disadvantage, and the consequences persist into adulthood. They encounter challenges in learning at school, have lower earning potential as adults, and face obstacles to engaging in their communities [3]. Most Tanzanians have undiversified diets, where an average of 71% of all energy comes from staple foods. Even in the wealthiest segment of the population, nearly 60% of energy is derived solely from staple foods [11]. The 2022 TDHS indicates that only 64% of children aged less than six months were exclusively breastfed, only 19% of children aged 6–23 months, and 25% of women aged 15–49 years received a diversified diet. In addition, 11% of children aged 6–59 months were given iron-containing supplements in the last 12 months at the health facility, and 53% were given vitamin A supplements in the last 6 months.

To the best of our knowledge, none of the previous studies conducted in Tanzania [4,6,10,12–14] have thoroughly examined the time trends of stunting alongside child-related factors (such as birth weight, gender, and age), maternal factors (including breastfeeding practices, maternal education, nutritional status, and the number of antenatal care visits), household-related factors (such as hygiene practices, socio-economic status, the gender of the household head, and child infections, particularly diarrhoea in the 2 weeks before the survey), and cross-regional comparisons from 1991/92 to 2022. Therefore, the primary aim of this study was to examine the trends of stunting and characteristics associated with the risk of being stunted in Tanzania.

## Materials and methods

### Ethical considerations

This is a secondary analysis of the Tanzania Demographic Health Surveys collected by the government through the National Bureau of Statistics. Prior to data collection, the protocols and data collection procedures were approved by the relevant authorities in mainland Tanzania and Zanzibar. These included the National Institute of Medical Research (NIMR), the Zanzibar Medical Research Ethical Committee (ZAMREC), the Institutional Review Board of ICF International, and the Centres for Disease Control and Prevention in Atlanta. All participants provided written informed consent before the interviews. For illiterate individuals, a written consent form was presented orally in the presence of a legally permitted representative or witness, such as a friend, relative, or someone not affiliated with the research team. Parental or guardian consent was obtained for children and adolescents participating in the study. Every qualified respondent was questioned in the most private setting possible, without the presence of a third party.

### Study design

The DHS are nationally representative, repeated cross-sectional surveys conducted in low- and middle-income countries. Seven waves of Tanzania DHS have been conducted since 1991−92–2022.

### Data sources

The Demographic and Health Surveys provide valuable information for comparing nutritional and health estimates within and across countries, monitoring trends, and measuring the progress of interventions targeting stunting among children. Among other measures, these gather socio-economic and demographic data on children under the age of five, along with other health and development measures. The resulting data contain both national and regional-level estimates for stunting indicators, essential for making inferences about national and regional trends. All data presented in the graphs as cross-sectional data points and tables are from the Tanzania DHS, 1991/92–2022 [15–21].

Anthropometry is commonly used to measure a child's nutritional status. Anthropometric measurements are used to report on child growth indicators. According to the World Health Organization (WHO) 2006 report, the distribution of height and weight among children under age 5 was compared with the WHO Child Growth Standards reference population [22].

The distribution of a well-nourished population will be similar to that of the reference population, while the distribution of a poorly nourished population will not. Children who are too short for their age are considered stunted, and this is the result of chronic or recurrent undernutrition. Stunting is the impaired growth and development that children experience due to poor nutrition, repeated infection, and inadequate psychosocial stimulation [23]. Children whose height-for-age z-score (HAZ) is below minus two standard deviations (−2 SD) from the median of the reference population are considered short for their age (stunted). Children whose HAZ is below minus three standard deviations (−3 SD) from the median are considered severely stunted [24,25]. HAZ is calculated by subtracting an age- and sex-appropriate median value from a standard population, and dividing it by the SD of the standard population.

The primary outcome of interest was stunting, defined as a height-for-age measurement that is more than two standard deviations below the mean of the reference population. In 2006, the WHO recommended new standard growth cut-off points for stunting [22]. The Tanzania DHS surveys conducted from 1991/92 to 2004/05 used the Center for Disease Control standard growth references, which were derived from the National Center for Health Statistics, Fels Research Institute, and Center for Disease Control reference population. To harmonize the surveys from 1991/2 to 2004/5, z-scores were recalculated according to the new WHO Child Growth Standards. The syntax file is available from the WHO (http://www.who.int/childgrowth/software/en/).

To maintaining the quality of measurements for the TDHS, quality control teams comprised personnel from the National Bureau of Statistics, the Office of the Chief Government Statistician, the Tanzania Food and Nutrition Centre (TFNC), and the ministry responsible for health in both mainland Tanzania and Zanzibar [26]. Monitoring of fieldwork involved regular visits to ensure the survey was conducted according to established protocols, providing real-time solutions to field challenges by observing the biomarker measurements of eligible respondents. All biomarker questionnaires and urine specimens were dispatched weekly to the nearest TFNC laboratory. Field check tables were regularly generated from Syncloud to monitor data quality and fieldwork progress. For field teams facing specific challenges, quality control staff provided targeted instructions to improve their performance; otherwise, consistent feedback was offered to all field teams.

### Data access

The use of this data was approved by the Demographic and Health Surveys Program, ICF, following our request with the data analysis concept. An authorisation letter was issued to the research team on 31 January 2024.

### TDHS sampling of households

The TDHS employed a random sampling method that took population density into account to gather data from all administrative regions of the country. Data of 42,408 under-five children had nutrition related variables. The seven waves of TDHS had different sample size, 1991–1992 (n = 6,585), 1996 (n = 5,438), 1999 (n = 2,555), 2004–2005 (n = 7,230), 2009–2010 (n = 6,792), 2015–2016 (n = 9,001), and 2022 (n = 4,807).

### Data analysis

**Descriptive analysis.** Descriptive statistics were used to summarize the data on stunting, while the factors influencing stunting (such as gender and age), caregiver/mother-related factors (such as education, number of ANC visits, and age at first birth), and household-level factors (such as gender of the household head and household wealth status) were analyzed using appropriate statistical methods for the Tanzania DHS 1991/92–2022 survey data. We calculated frequencies and percentages, which are displayed using graphs as cross-sectional data points. A chi-square test was used to assess the association between stunting and each of the independent variables for identifying factors associated with stunting. The descriptive statistics guided the selection of covariates included in the univariate and multivariable logistic regression model to assess the factors influencing stunting in 2022.

**Inferential analysis.** It has been hypothesised that child stunting would be affected by both child, caregivers, and household-level factors. A multivariable logistic regression model was estimated to identify factors associated with stunting in 2022. Both backward and forward elimination have been used to arrive at the final model for statistically significant levels, conditional on the p-value being less than 5 percent ($p < 0.05$). All analyses have been weighted using the weighting generated by the TDHS. STATA (StataCorp, College Station, TX, USA) version 15 was used for data management and analysis. The 'svy' commands were utilised to account for the cluster sampling design, sampling weights, and the calculation of standard errors.

## Results

### Profile of the respondents 1991-92 to 2022

Table 1 presents information on the profile of the respondents from the repeated Tanzania DHS cross-sectional surveys conducted from 1991-92 to 2022. The findings indicated that across all seven waves of the Tanzania DHS, there was minimal variation in the proportion of male and female children included in the studies. Close to half of children represented in all surveys were aged between 18 and 47 months, ranging between 47.9% and 49.5% for all TDHS waves.

Maternal characteristics demonstrate a positive trend in educational attainment among caregivers. Over the past 30 years, the percentage of caregivers without any formal education has decreased from 37.8% in the 1991–1992 period to 21.5% in the 2022 survey. Similarly, the proportion of caregivers with secondary education or higher has risen from 3.3% in 1991–1992 to 26% in the 2022 survey. Across all the Tanzania DHS, the majority (> 63.1%) of caregivers had a normal weight (Body Mass Index [kg/m²] – 18.5 to 24.9).

Household-level characteristics indicate that the majority of household heads were male across all surveys, with proportions ranging from 77.2% to 88.4%. Additionally, most respondents in all surveys were sampled from rural areas, with percentages ranging from 73.2% to 84.7%.

### Trends of stunting in Tanzania

In Tanzania, stunting has decreased slowly over the last 30 years, from 50% in 1991/92–30% in 2022 (Fig 1).

### Characteristics associated with the risk of stunting (child, mother, and family)

**Stunting and child gender.** Fig 2 shows this disparity over time in Tanzania, where, over the last 30 years, under-five boys have consistently had 4–7 percentage points (ppts) higher stunting rates than girls of the same age. Girls have a 16% lower chance of being stunted.

**Stunting and child age.** Fig 3 highlights this pattern in Tanzania, where, until approximately two years, stunting rates increase and decrease thereafter. Stunting rates are currently at their highest in children of 18–24 months (40%) and 24–30 months (41%).

**Stunting and child birthweight.** Fig 4 shows the relationship between birthweight and stunting over time in Tanzania. Close to half of children under five years old who were born with low birthweight (<2,500 grams) are stunted, and only around a quarter of children who weighted 3,500 grams or more at birth.

### Maternal characteristics associated with the risk of stunting

**Breastfeeding practices.** There has been a decline in the stunting trend for those breastfeeding for more than 12 months in Tanzania, from 51% in 1991/92 to 38% in the 1996 DHS, while stunting rates for those breastfeeding for less than 12 months declined by 5 ppts from 22% in 1991/92–17% in 2022 (Fig 5).

**Table 1. Characteristics of children, caregivers, and households from 1991−92 to 2022.**

| | TDHS 1991−92 N=6,585 | TDHS 1996 N=5,438 | TDHS 1999 N=2,555 | TDHS 2004−05 N=7,230 | TDHS 2010 N=6,792 | TDHS 2015−16 N=9,001 | TDHS 2020 N=4,807 |
|---|---|---|---|---|---|---|---|
| | n(%) | n(%) | n(%) | n(%) | n(%) | n(%) | n(%) |
| Child Characteristics | | | | | | | |
| Child sex | | | | | | | |
| Male | 3,254(49.4) | 2,760(50.7) | 1,290(50.5) | 3,609(49.9) | 3,374(49.7) | 4,508(50.1) | 2,426(50.5) |
| Female | 3,331(50.6) | 2,678(49.3) | 1,265(49.5) | 3,621(50.1) | 3,418(50.3) | 4,493(49.9) | 2,381(49.5) |
| Child age, months | | | | | | | |
| <9 | 1,154(17.5) | 949(17.4) | 466(18.2) | 1,253(17.3) | 1,107(16.3) | 1,472(16.5) | 760(15.8) |
| 9 - 17 | 1,180(17.9) | 968(17.7) | 419(16.4) | 1,264(17.5) | 1,118(16.5) | 1,560(17.5) | 775(16.1) |
| 18–47 | 3,155(47.9) | 2,607(47.9) | 1,227(48.0) | 3,463(47.9) | 3,361(49.5) | 4,324(48.4) | 2,365(49.2) |
| 48–59 | 1,096(16.6) | 914(16.8) | 443(17.3) | 1,250(17.3) | 1,206(17.7) | 1,575(17.6) | 876(18.2) |
| Maternal characteristics | | | | | | | |
| Mother's Education | | | | | | | |
| None | 2,488(37.8) | 1,586(29.2) | 706(27.6) | 1,982(27.4) | 1,735(25.5) | 1,964(21.8) | 1,033(21.5) |
| Primary | 3,882(58.9) | 3,582(66.0) | 1,546(60.5) | 4,641(64.2) | 4,281(63.0) | 5,411(60.1) | 2,522(52.5) |
| Secondary and above | 215(3.3) | 258(4.7) | 303(11.8) | 607(8.4) | 776(11.4) | 1,626(18.1) | 1,252(26.0) |
| Body mass index (kg/m2) of mother | | | | | | | |
| Underweight (<18.50) | 610(9.3) | 472(8.7) | | 641(8.9) | 664(9.8) | 642(7.1) | 332(6.9) |
| Normal (18.50–24.99) | 5,213(79.5) | 4,197(77.9) | | 5481(76.1) | 4,787(70.7) | 6,057(67.5) | 3,031(63.1) |
| Overweight (>=25.0) | 731(11.1) | 717(13.3) | | 1,085(15.0) | 1321(19.5) | 2,280(25.4) | 1,435(29.8) |
| ANC visits | | | | | | | |
| < 4 visits | 2,165(32.9) | 1,614(29.7) | 1,301(50.9) | 4,111(56.8) | 4,735(69.7) | 5,828(64.7) | 3,043(63.3) |
| >= 4 visits | 4,420(67.1) | 3,824(70.3) | 1,254(49.1) | 3,119(43.1) | 2,057(30.3) | 3,173(35.3) | 1,764(36.7) |
| Household Related Factors | | | | | | | |
| Gender of Head of Household | | | | | | | |
| Female | 762(11.6) | 842(15.5) | 395(15.5) | 6,075(15.9) | 1,094(16.1) | 1,420(15.8) | 1,094(22.8) |
| Male | 5,823(88.4) | 4,596(84.5) | 2,160(84.5) | 1,155(84.0) | 5,698(83.9) | 7,581(84.2) | 3,713(77.2) |
| Locality | | | | | | | |
| Rural | 5,575(84.7) | 4,376(80.5) | 1,909(74.7) | 6,001(83.0) | 5,554(81.8) | 6,966(77.4) | 3,517(73.2) |
| Urban | 1,010(15.3) | 1,062(19.5) | 646(25.3) | 1,229(17.0) | 1,238(18.2) | 2,035(22.6) | 1,290(26.8) |
| SES | | | | | | | |
| Lowest | 1,952(30,4) | 1,544(28.4) | 930(36.4) | 1,590(21.9) | 1,389(20.4) | 2,080(23.1) | 1,029(21.4) |
| Second | 615(9.6) | 654(12.0) | 211(8.3) | 1,453(20.1) | 1,561(22.9) | 1,879(20.9) | 951(19.8) |
| Middle | 1,320(20.6) | 1,469(27.0) | 686(26.8) | 1,463(20.2) | 1,453(21.4) | 1,756(19.5) | 993(20.7) |
| Fourth | 1,240(19.5) | 698(12.8) | 240(9.4) | 1,605(22.2) | 1,393(20.5) | 1,854(20.6) | 976(20.3) |
| Highest | 1,278(19.9) | 1,073(19.7) | 480(19.1) | 1,119(15.5) | 996(14.7) | 1,432(15.9) | 858(17.8) |

**Mother's education.** Fig 6 shows the trend of stunting and maternal education in Tanzania. It is observed that mothers with no education have children with higher rates of stunting compared to those with higher education (primary complete). There has been a declining trend since the 1996 DHS for all education categories. For example, stunting among caregivers with no education declined by 11 ppts from 47% to 36% in the 2022 DHS.

**Stunting and Mother's nutritional status.** Mother's nutritional characteristics in Tanzania across the years have been consistently closely correlated with the growth of the newborn (Fig 7). Mothers with a low body mass index have a higher proportion of stunted children.

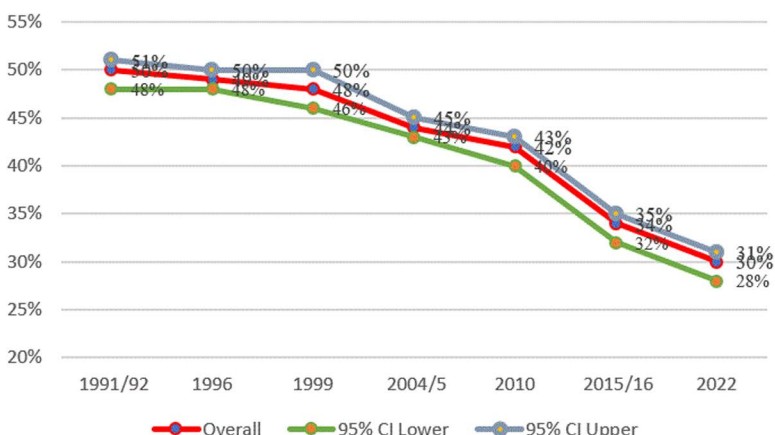

**Fig 1. Prevalence of stunting from Tanzania Demographic Health Surveys 1991–2022.** The red line shows the stunting rates, which is displaying a declining trend. The line graphs represent cross-sectional data points for different TDHS from 1991/92–2022, rather than longitudinal data for all the graphs.

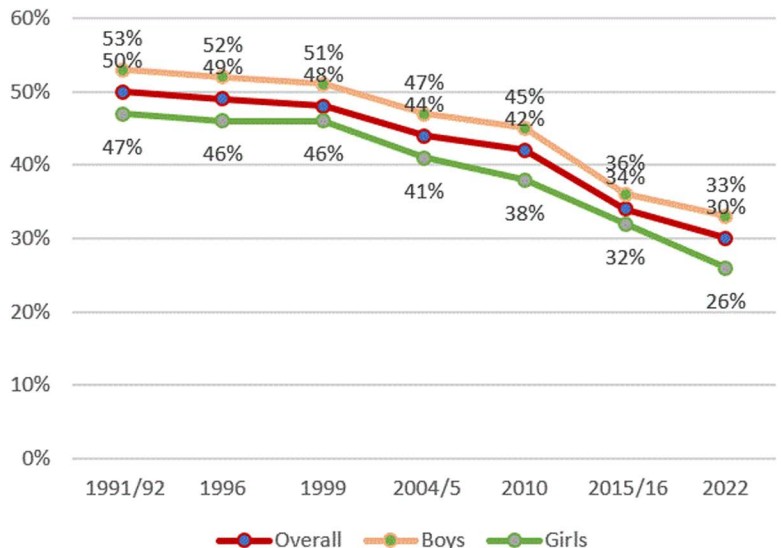

**Fig 2. Prevalence of stunting and child gender from TDHS 1991/92–2022.** The red line represents the overall stunting rates, the orange line represents stunting rates among boys, and the green line represents stunting rates among girls.

**Stunting and number of ANC visits.** Fig 8 shows the trend of the ANC visits over time. An increased number of ANC visits to more than 4 has been associated with a declining trend of stunting. A 16-point decline in stunting was observed for mothers having more than 4 ANC visits, from 42% in 1991−92 to 26% in 2022.

## Household related factors

**Stunting and access to improved water.** Fig 9 shows the trends in stunting and access to improved water. Stunting rates among households with access to unimproved water are higher than at the national level.

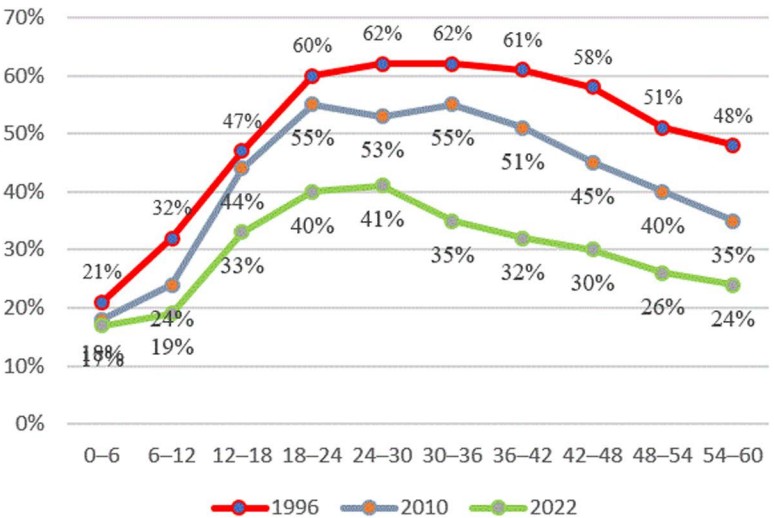

**Fig 3. Prevalence of stunting and age in months from TDHS from 1991/92–2022. The red line represents the stunting rates for 1996, the grey line represents stunting rates for 2010, and the green line represents stunting rates for 2022.**

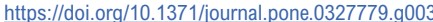

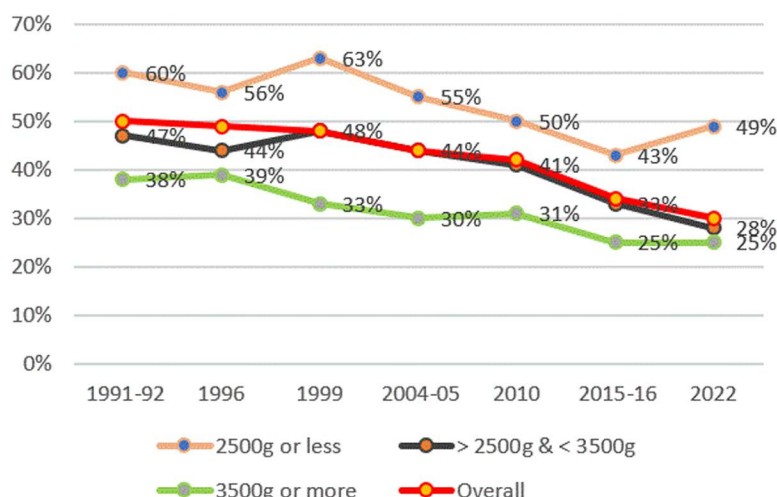

**Fig 4. Prevalence of stunting and birthweight from TDHS 1991/92–2022. The red line represents overall stunting rates, black line for those with weights>2,500 grams &<3,500 grams, the orange line represents stunting rates for those with weights less than 2,500 grams, and the green line represents stunting rates for those with weights above 3,500 grams.**

**Stunting and hygiene practices.** Fig 10 shows the trends in stunting and household sanitation facilities. Stunting rates among households with unimproved sanitation and open defecation are higher than at the national level.

**Stunting and Household Socioeconomic Status (SES).** Fig 11 shows a clear disparity in stunting rates between households of different socioeconomic statuses (SES), with households in the highest wealth quintile having lower stunting rates than those in the lowest wealth quintile.

**Stunting and gender of head of household.** There is a declining stunting trend within male-headed households from 43% in 1991 to 30% in 2022. The proportion of stunting within female-headed households has been higher than that of male-headed households from 1991−2 to 2015−16 (Fig 12).

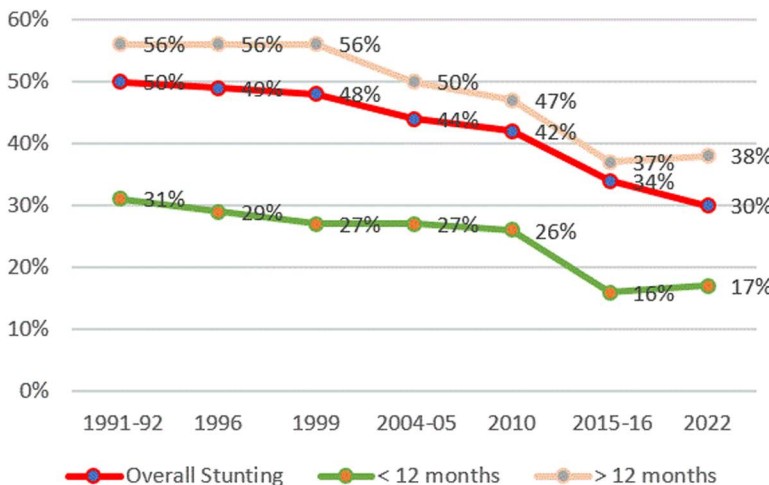

**Fig 5. Prevalence of stunting and breastfeeding duration from TDHS 1991/92 to 2022.** The red line represents the overall stunting rates, the orange line represents stunting rates for those who maintained breastfeeding for more than one year and above, the green line represents stunting rates for those with less than 12 months breastfeeding.

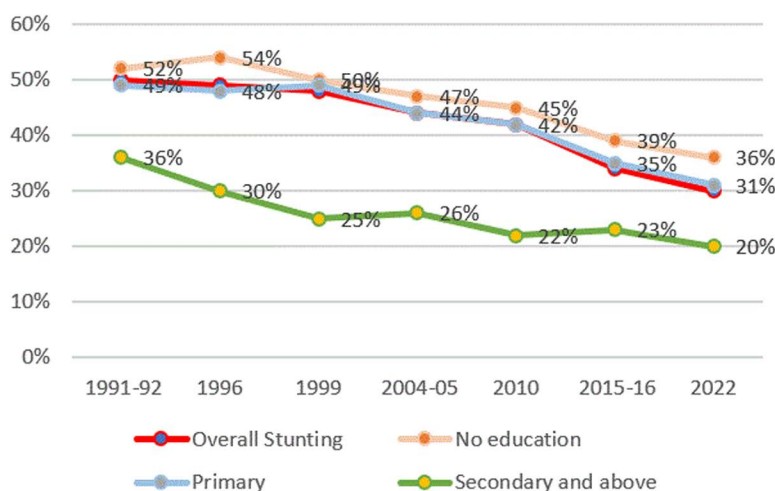

**Fig 6. Prevalence of stunting and mother's education from TDHS 1991/92–2022.** The red line represents the overall stunting rates, the green line represents stunting rates for those who attained secondary education and above, the grey line represents stunting rates for those with primary education, and the orange line represents those who had not attended formal education.

**Stunting and child infection (Diarrhoea).** The prevalence of diarrhoea among children under 5 declined from 31% and 13% in 1991 to 8% and 9% respectively in 2022 in the 2 weeks before the survey. It has been observed that as diarrhoea declines over the years, stunting also decreases. Trends of stunting and diarrhoea are presented in Fig 13. Repeated episodes of sickness deteriorate a child's growth over time.

## Stunting by regions

Fig 14 presents information on time trends in stunting in a few selected regions that were performing relatively well in 2022 (Dar es Salaam, Pwani, Kilimanjaro). Interestingly, whereas Dar es Salaam and Kilimanjaro have consistently been

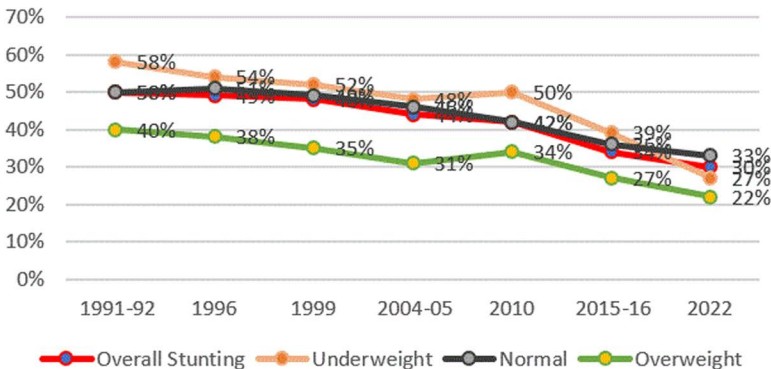

**Fig 7. Prevalence of stunting and mothers' nutritional status from TDHS 1991/92 to 2022.** The red line represents the overall stunting rates, the black line presents stunting rates for mothers who have normal weight, the orange line represents stunting rates for overweight mothers, and the green line represents stunting rates for underweight mothers.

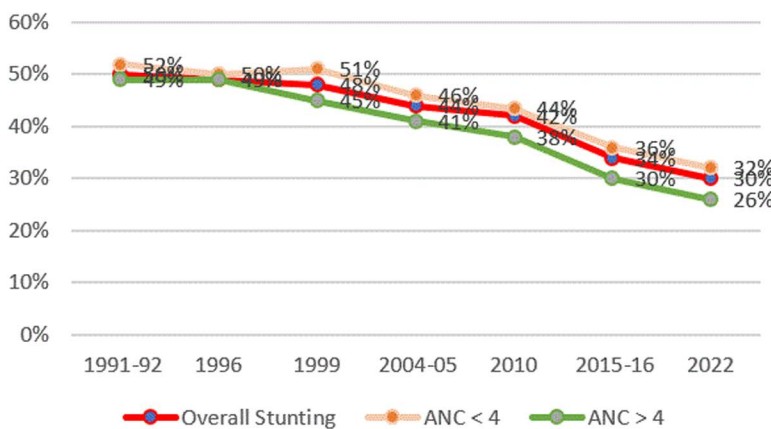

**Fig 8. Prevalence of stunting and ANC visits from TDHS 1991/92–2022.** The red line represents the overall stunting rates, the green line presents stunting rates for mothers who had more than 4 ANC visits, and the orange line represents stunting rates for mothers who had less than 4 ANC visits.

outperforming the average region in Tanzania from 1992–2022 in terms of stunting rates, Pwani started off as a relatively poor performer in 1992 but over time has become a relatively well-performing region.

Fig 15, in turn, presents information on trends in stunting in a few selected regions that were performing poorly in 2022 (Iringa, Ruvuma, and Mbeya). Note that all three regions have been performing relatively poorly ever since 1992.

### Bivariate analysis of childhood stunting and related factors in 2022

Table 2 presents bivariate analysis relating 2022 stunting rates to various characteristics associated with the risk of being stunted, including demographic characteristics such as age, place of residence, child gender, zone, mother's educational status, and wealth index quintile. The South West Highlands and Southern Highlands regions had a statistically significantly higher proportion of stunted children, at 37.8% and 46.2% respectively, compared to Eastern and Zanzibar, which had 22.8% and 17.1% respectively. A higher proportion of stunted children was observed among those aged 18 to 47 months (35.9%) and 9 to 17 months (27.7%). Similarly, males had a higher proportion (33.3%) of stunting compared to

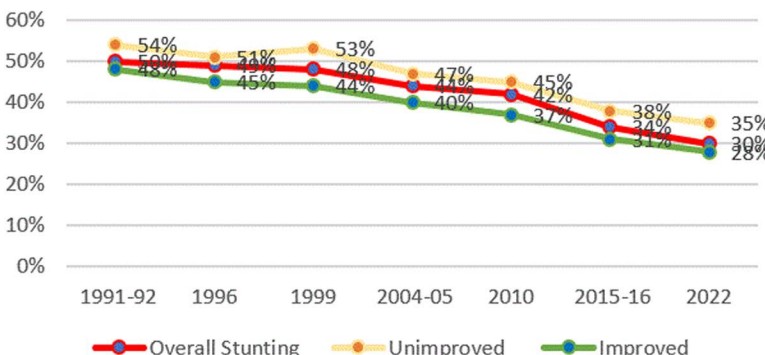

**Fig 9. Prevalence of stunting and household access to water from TDHS from 1991/92 to 2022. The red line represents the overall stunting rates, the green line presents stunting rates for those with access to improved water, and the orange line represents stunting rates for those without access to improved water.**

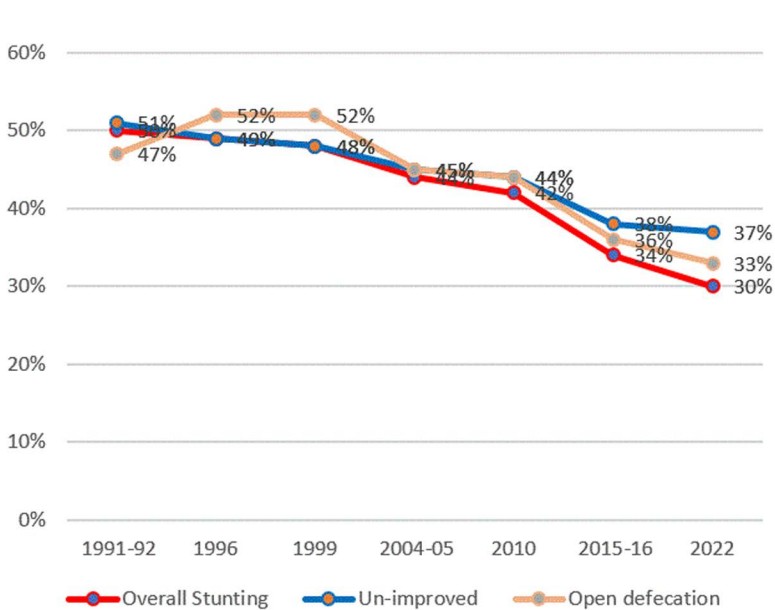

**Fig 10. Prevalence of stunting and household access to sanitation facilities in Tanzania from the TDHS 1991/92 to 2022. The red line represents the average stunting rates, the blue line represents stunting rates for households with unimproved sanitation facilities, and the orange line represents stunting rates for households practicing open defecation.**

females (25.8%). A high proportion (55.6%) of stunting was observed in children with low birthweight. Women with short birth intervals (less than 24 months) had a higher proportion (35.3%) of stunted children compared to those with intervals of 24 to 47 months (29.5%) and those above 47 months (25.6%). Overweight and obese women had a lower proportion of stunted children (24.5% each) compared to those with normal weight (32.9%). Women with secondary education or above had a lower proportion of stunted children (20.3%) compared to those with no formal education (36.3%). Those with more than four ANC visits had a lower proportion of stunted children (26.5%). Women who gave birth before the age of 20 had a higher proportion of stunted children; similarly, those exposed to alcohol had a higher proportion of stunted children (38.3%). Women who used folic acid for more than 90 days had a lower proportion of stunted children (27.8%). Households with limited access to improved water had a higher proportion of stunted children (34.9%); similarly, those lacking

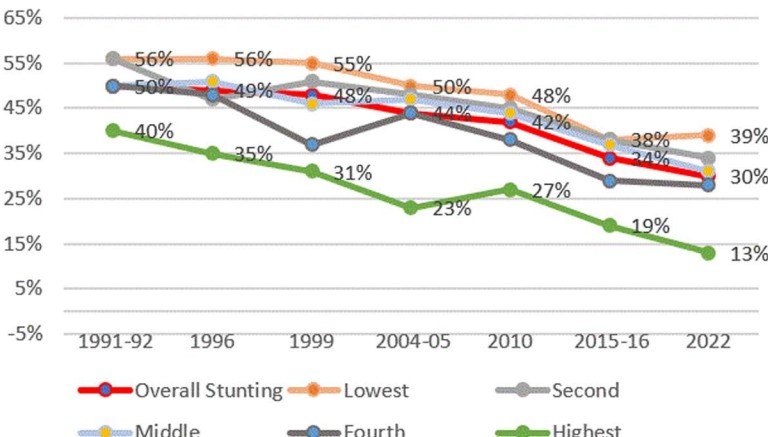

**Fig 11. Prevalence of stunting and household ranking for each person using socioeconomic status from TDHS 1991/92 to 2022.** The red line represents the average stunting rates, the orange line represents stunting rates for households within the lowest SES, and the green line represents stunting rates for households within the highest SES.

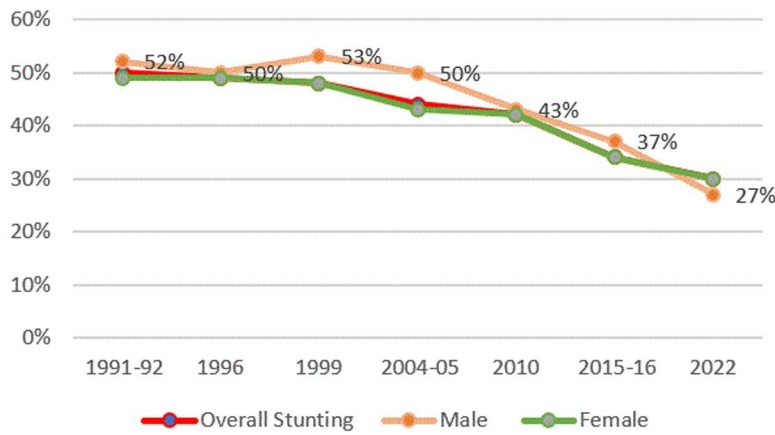

**Fig 12. Prevalence of stunting and male vs female household head from TDHS 1991/92 to 2022.** The red line represents the average stunting rates, the green line represents stunting rates for female-headed, the orange line represents stunting rates for male headed households.

access to improved sanitation facilities (37.1%). Households in the lowest SES had the highest proportion of stunted children (38.6%) compared to those in the highest SES (12.6%).

## Multivariable logistic regression on the characteristics associated with the risk of stunting in 2022

A total of 4,807 observations were included in the regression analysis (Table 3). The table presents the results of a multivariable logistic regression analysis on factors associated with stunting in Tanzania.

Several factors in the multivariable analysis remain significantly associated with stunting even after controlling for confounding factors. These significant variables include households residing in the Southern Highlands (p<0.001); child age (p<0.001), child sex (p<0.001), child birth weight (p<0.001), birth interval (p<0.05), mother's (p<0.05), mother's education (p<0.05), delivery by Caesarean section (p<0.05), access to sanitation (p<0.05), and socio-economic status (SES) (p<0.001).

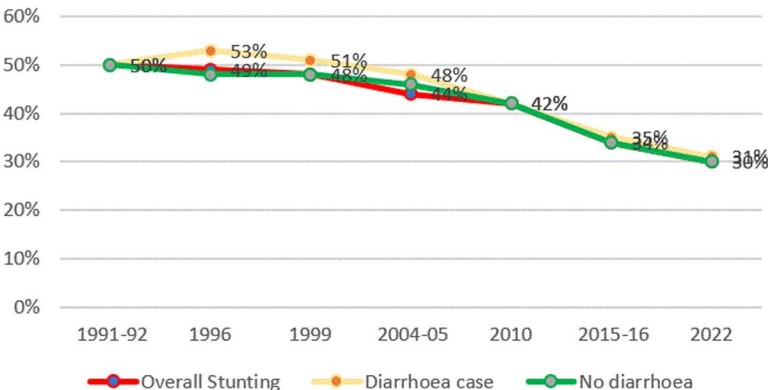

**Fig 13. Prevalence of stunting and child diarrhoea from TDHS 1991/92 to 2022. The red line represents the average stunting rates, the green line represents stunting rates for child who had no diarrhoea, and the orange line represents stunting rates for child who had diarrhoea.**

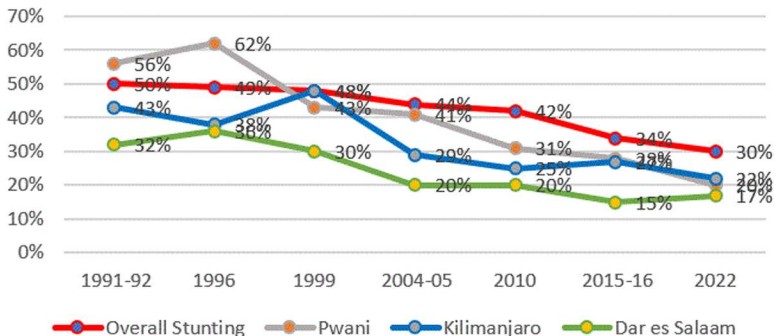

**Fig 14. Prevalence of stunting and regions with low stunting rates from TDHS 1991/92–2022. The red line represents the average stunting rates, the green line represents stunting rates for the Dar es Salaam region, the blue line represents stunting rates for the Kilimanjaro region, and the grey line represents stunting rates for the Pwani region.**

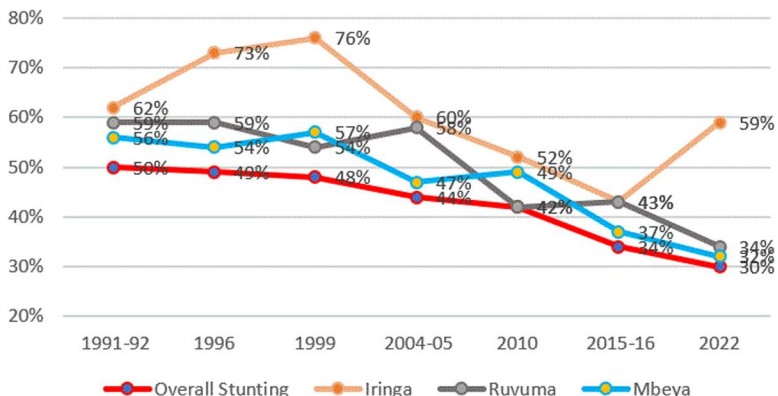

**Fig 15. Prevalence of stunting and regions with high stunting rates from TDHS 1991/92–2022. The red line represents the average stunting rates, the orange line represents stunting rates for the Iringa region, the grey line represents stunting rates for the Ruvuma region, and the blue line represents stunting rates for the Mbeya region.**

**Table 2. Bivariate Analysis of Childhood Stunting and Related Factors, TDHS 2022.**

| | Not Stunted n = 3,423 | Stunted n = 1,443 | χ2 (df) | P-value |
|---|---|---|---|---|
| Zones new | n(%) | n(%) | | |
| Lakes | 1,197(69.7) | 520(30.3) | 109.6(8) | <0.001 |
| Northern highlands | 369(74.0) | 129(25.9) | | |
| Central | 363(70.1) | 151(29.3) | | |
| Western | 328(70.3) | 138(29.7) | | |
| South West Highlands | 296(62.1) | 181(37.9) | | |
| Southern Highlands | 136(53.8) | 117(46.2) | | |
| Eastern | 471(77.2) | 139(22.8) | | |
| Southern | 143(77.1) | 43(22.9) | | |
| Zanzibar | 118(82.9) | 24(17.1) | | |
| Child age, months | | | | |
| <9 | 615(82.7) | 129(17.3) | 81.5(4) | <0.001 |
| 9 - 17 | 584(72.3) | 224(27.7) | | |
| 18–47 | 1,527(64.1) | 855(35.9) | | |
| 48–59 | 670(75.2) | 221(24.8) | | |
| Child sex | | | | |
| Male | 1,653(66.7) | 826(33.3) | 18.2(1) | <0.001 |
| Female | 1,769(74.2) | 617(25.8) | | |
| Child weight | | | | |
| Low birth weight | 82(44.4) | 103(55.6) | 52.3(2) | <0.001 |
| Normal | 1,628(72.4) | 619(27.6) | | |
| Child fever | | | | |
| Yes | 376(70.1) | 161(29.9) | 0.17(1) | 0.679 |
| No | 3,046(70.4) | 1,282(29.6) | | |
| Child Diarrhoea in the 2 weeks before the survey | | | | |
| Yes | 294(69.6) | 129(30.5) | 0.09(1) | 0.755 |
| No | 3,129(70.4) | 1,314(29.6) | | |
| Birth order | | | | |
| 1 | 745(68.6) | 341(31.4) | 16.5(3) | <0.001 |
| 2-3 | 1,347(72.2) | 519(27.8) | | |
| 4-5 | 796(73.9) | 280(26.0) | | |
| Six and above | 535(63.9) | 302(36.1) | | |
| Birth interval | | | | |
| < 24 months | 398(64.7) | 217(35.3) | 14.6(3) | 0.002 |
| 24 - 47 months | 1,322(70.5) | 554(29.5) | | |
| Above 47 months | 954(74.4) | 328(25.6) | | |
| BMI | | | | |
| Underweight | 258(72.9) | 96(27.1) | 47.6(4) | <0.001 |
| Normal | 2,143(67.1) | 1,051(32.9) | | |
| Overweight | 660(75.5) | 214(24.5) | | |
| Obese | 350(81.8) | 213(24.5) | | |
| Education | | | | |
| None | 667(63.7) | 380(36.3) | 60.9(2) | <0.001 |
| Primary | 1,955(69.5) | 859(30.5) | | |
| Secondary and above | 800(79.7) | 203(20.3) | | |
| ANC visits | | | | |

*(Continued)*

**Table 2.** (Continued)

| | Not Stunted n = 3,423 | Stunted n = 1,443 | χ2 (df) | P-value |
|---|---|---|---|---|
| < 4 visits | 2,108(68.5) | 970(31.5) | 13.8(1) | <0.001 |
| >= 4 visits | 1,314(73.5) | 473(26.5) | | |
| Delivery – caesarean section | | | | |
| No | 3,221(70.5) | 1,348(29.5) | 2.3(1) | 0.129 |
| Yes | 202(67.9) | 95(32.0) | | |
| Age at birth | | | | |
| <20 years | 1,983(68.1) | 929(31.9) | 8.7(2) | 0.013 |
| 20-24 | 1,137(73.2) | 416(26.8) | | |
| 25+ | 302(75.7) | 97(24.3) | | |
| Exposure to alcohol | | | | |
| No | 3,286(70.7) | 1,358(29.2) | 8.3(1) | 0.004 |
| Yes | 136(61.7) | 85(38.3) | | |
| Used folic acid for more than 90 days | | | | |
| No | 2,628(69.8) | 1,137(30.2) | 3.6(1) | 0.058 |
| Yes | 794(72.2) | 305(27.8) | | |
| Access to improved water | | | | |
| No | 953(65.0) | 512(34.9) | 34.9(2) | <0.001 |
| Yes | 2,350(72.4) | 894(27.6) | | |
| Assess to improved sanitation | | | | |
| Unimproved | 1,244(62.9) | 734(37.1) | 113.4(3) | <0.001 |
| Improved | 1,686(77.4) | 492(22.6) | | |
| Open defecation | 389(67.3) | 189(32.8) | | |
| Household head | | | | |
| Female | 805(72.7) | 302(27.3) | 4.1(1) | 0.044 |
| Male | 2,618(69.6) | 1,140(30.3) | | |
| SES | | | | |
| Lowest | 687(61.5) | 430(38.6) | 139.7(4) | <0.001 |
| Second | 633(61.5) | 333(34.5) | | |
| Middle | 659(65.5) | 295(30.9) | | |
| Fourth | 734(72.3) | 281(27.7) | | |
| Highest | 709(87.4) | 102(12.6) | | |

Note: Regions belonging to the South West Highlands include Katavi, Rukwa, Songwe, and Mbeya, while the Southern Highlands comprise Iringa, Njombe, and Ruvuma.

Factors such as child diarrhoea (AOR = 1.215 [95% CI: 0.899–1.643], p = 0.204), more than four ANC visits (AOR = 0.858 [95% CI: 0.694–1.061], p = 0.159), mother's age at birth of 20–24 years (AOR = 0.901 [95% CI: 0.750–1.081], p = 0.260), mothers above 25 years of age (AOR = 0.998 [95% CI: 0.742–1.341], p = 0.988), and household access to improved water (AOR = 1.012 [95% CI: 0.842–1.217], p = 0.896) were not significantly related to child stunting.

## Discussion

This study sought to examine trends in stunting among children under five in mainland Tanzania and to identify factors associated with stunting in this population. Despite the persistent high burden of stunting, a notable decline has been

**Table 3. Multivariable logistic regression on the characteristics associated with the risk of stunting, TDHS 2022.**

| | COR | 95%CI | P-value | AOR | 95%CI | P-value |
|---|---|---|---|---|---|---|
| Zones new | | | | | | |
| Lakes | 1 | | | | | |
| Northern highlands | 0.807 | 0.602-1.083 | 0.153 | 0.916 | 0.673-1.246 | 0.576 |
| Central | 0.953 | 0.728-1.247 | 0.729 | 0.931 | 0.696-1.245 | 0.629 |
| Western | 0.975 | 0.744-1.277 | 0.853 | 0.787 | 0.592-1.045 | 0.098 |
| South West Highlands | 1.407 | 1.115-1.776 | 0.004 | 1.408 | 1.104-1.794 | 0.006 |
| Southern Highlands | 1.977 | 1.510-2.589 | <0.001 | 2.368 | 1.746-3.212 | <0.001 |
| Eastern | 0.681 | 0.511-0.905 | 0.008 | 0.961 | 0.702-1.315 | 0.804 |
| Southern | 0.683 | 0.453-1.030 | 0.069 | 0.612 | 0.392-0.956 | 0.031 |
| Zanzibar | 0.474 | 0.365-0.616 | <0.001 | 0.894 | 0.665-1.202 | 0.459 |
| Child age, months | | | | | | |
| <9 | 1 | | | | | |
| 9 - 17 | 1.831 | 1.356-2.470 | <0.001 | 1.866 | 1.351-2.578 | <0.001 |
| 18 - 47 | 2.678 | 2.072-3.460 | <0.001 | 3.176 | 2.381-4.237 | <0.001 |
| 48 - 59 | 1.574 | 1.172-2.113 | 0.003 | 2.037 | 1.416-2.931 | <0.001 |
| Child sex | | | | | | |
| Female | 1 | | | | | |
| Male | 1.434 | 1.229-1.673 | <0.001 | 1.584 | 1.349-1.858 | <0.001 |
| Child weight | | | | | | |
| Normal | 1 | | | | | |
| Low birth weight | 3.297 | 2.197-4.947 | <0.001 | 3.639 | 2.279-5.810 | <0.001 |
| Child fever | | | | | | |
| No | 1 | | | | | |
| Yes | 1.015 | 0.776-1.328 | 0.913 | | | |
| Child Diarrhoea in the 2 weeks before the survey | | | | | | |
| No | 1 | | | 1 | | |
| Yes | 1.042 | 0.790-1.374 | 0.769 | 1.215 | 0.899-1.643 | 0.204 |
| Birth order | | | | | | |
| <2 | 1 | | | | | |
| 2-3 | 0.842 | 0.684-1.037 | 0.107 | 0.747 | 0.093-5.976 | 0.783 |
| 4-5 | 0.769 | 0.608-0.971 | 0.027 | 0.577 | 0.071-4.664 | 0.606 |
| Six and above | 1.233 | 0.973-1.563 | 0.083 | 0.839 | 0.104-6.771 | 0.869 |
| Birth interval | | | | | | |
| < 24 months | 1 | | | | | |
| 24 - 47 months | 0.769 | 0.605-0.976 | 0.031 | 0.762 | 0.598-0.969 | 0.027 |
| Above 47 months | 0.631 | 0.489-0.814 | <0.001 | 0.721 | 0.574-1.000 | 0.051 |
| BMI | | | | | | |
| Normal | 1 | | | | | |
| Underweight | 0.758 | 0.552-1.041 | 0.086 | 0.720 | 0.520-0.997 | 0.048 |
| Overweight | 0.660 | 0.537-0.811 | <0.001 | 0.803 | 0.641-1.005 | 0.055 |
| Obese | 0.453 | 0.176-0.612 | <0.001 | 0.738 | 0.529-1.026 | 0.071 |
| Education | | | | | | |
| None | 1 | | | | | |
| Primary | 0.772 | 0.641-0.931 | 0.007 | 0.933 | 0.757-1.148 | 0.513 |
| Secondary and above | 0.446 | 0.352-0.566 | <0.001 | 0.715 | 0.530-0.963 | 0.027 |
| ANC visits and above | | | | | | |

*(Continued)*

**Table 3.** (Continued)

| | COR | 95%CI | P-value | AOR | 95%CI | P-value |
|---|---|---|---|---|---|---|
| < 4 visits | 1 | | | | | |
| >= 4 visits | 0.781 | 0.661-0.923 | 0.004 | 0.858 | 0.694-1.061 | 0.159 |
| Delivery Caesarean section | | | | | | |
| No | 0 | | | | | |
| Yes | 1.127 | 0.856-1.483 | 0.395 | 1.449 | 1.046-2.007 | 0.025 |
| Age at birth | | | | | | |
| <20 years | 1 | | | | | |
| 20-24 | 0.782 | 0.661-0.926 | 0.004 | 0.901 | 0.751-1.081 | 0.260 |
| 25+ | 0.685 | 0.522-0.897 | 0.006 | 0.997 | 0.742-1.341 | 0.988 |
| Exposure to alcohol | | | | | | |
| No | 1 | | | | | |
| Yes | 1.394 | 1.034-1.879 | 0.029 | 1.441 | 1.057-1.963 | 0.021 |
| Used folic acid for more than 90 days | | | | | | |
| No | 1 | | | | | |
| Yes | 0.889 | 0.733-1.079 | 0.236 | | | |
| Access to improved water | | | | | | |
| No | 1 | | | 1 | | |
| Yes | 0.709 | 0.602-0.832 | <0.001 | 1.012 | 0.842-1.217 | 0.896 |
| Assess to sanitation | | | | | | |
| Improved | 1 | | | 1 | | |
| Unimproved | 2.019 | 1.697-2.404 | <0.001 | 1.345 | 1.055-1.714 | 0.017 |
| Open defecation | 1.662 | 1.299-2.125 | <0.001 | 1.006 | 0.723-1.402 | 0.969 |
| Household head | | | | | | |
| Female | 1 | | | | | |
| Male | 1.158 | 0.961-1.396 | 0.124 | | | |
| SES | | | | | | |
| Lowest | 1 | | | | | |
| Second | 0.839 | 0.686-1.026 | 0.088 | 0.739 | 0.590-0.927 | 0.009 |
| Middle | 0.714 | 0.572-0.891 | 0.003 | 0.688 | 0.524-0.905 | 0.007 |
| Fourth | 0.611 | 0.477-0.784 | <0.001 | 0.628 | 0.446-0.883 | 0.008 |
| Highest | 0.230 | 0.174-0.305 | <0.001 | 0.268 | 0.178-0.403 | <0.001 |
| Observations | | | | 4,807 | | |
| Wald chi2 (39) | | | | 337.6 | | |
| Prob > chi2 | | | | <0.001 | | |
| Pseudo R2 | | | | 0.098 | | |

Note: Both backward and forward elimination have been used to arrive at the final multivariable model for statistically significant levels, conditional on the p-value being greater than 5 percent (p > 0.05).

observed — from 50% in 1991/2 to 30% in 2022 — highlighting progress amongst ongoing challenges. Several maternal, child, household, and contextual factors remain significantly associated with stunting in 2022.

Maternal factors. Consistent with existing literature, our findings reinforce the importance of birth spacing—specifically, that increased intervals between births significantly reduce the risk of child stunting. Short interpregnancy intervals are linked to adverse outcomes such as preterm birth, low birth weight, and congenital abnormalities [27–29]. Additionally, maternal nutritional status, as indicated by BMI, plays a vital role: higher maternal BMI correlates with lower stunting risk.

This aligns with evidence emphasizing the importance of adequate maternal nutrition during preconception and pregnancy for optimal fetal development [30–36], given that maternal undernutrition increases risks such as intrauterine growth restriction [37,38]. For instance, Amaha et al. (2021) demonstrated that each unit increase in maternal BMI reduces the odds of child stunting by approximately 4 percentage points [39]. Similar trends are observed within Tanzania, where children of undernourished mothers (BMI < 18.5) are more likely to be underweight [6]. These findings underscore the impact of maternal nutritional support and care on child growth outcomes.

Globally and within Tanzania, maternal education remains a consistent predictor of child nutritional status. Higher levels of maternal education are associated with reduced stunting prevalence, likely due to improved health literacy, better feeding practices, and increased utilization of healthcare services [40–47]. Mohamed (2023) attributes lower stunting rates among children of educated mothers to their increased awareness of nutrition and participation in growth monitoring programs [48]. Similarly, Moshi (2022) reports that maternal education predicts better child weight outcomes [6]. Education enhances household income and caregivers' capacity to provide appropriate nutrition, hygiene, and healthcare interventions [42,49].

The 2018 National Antenatal Care Guidelines recommend at least eight ANC visits based on the 2016 WHO model. Our findings indicate that attending four or more ANC visits and delivering in a health facility are associated with decreased rates of severe stunting, aligning with prior research [34–36,39]. For example, a Northern Tanzania study revealed that women attending fewer than four ANC visits had higher risks of delivering low birth weight infants, a key driver of stunting [61].

An intriguing observation is that, over time, children breastfed for more than 12 months consistently exhibit higher stunting prevalence compared to the overall rate, a pattern echoed in other regional studies. Akombi et al. (2017) identified prolonged breastfeeding beyond 12 months as a contributing factor to stunting in sub-Saharan Africa [50], and Cetthakrikul et al. (2020) found similar associations intensified by household economic hardship [51]. The complex relationship between breastfeeding duration and nutritional outcomes warrants further examination, considering the potential confounding effect of socioeconomic factors. One assumption could be that women who breastfeed for a longer period delay or substitute complementary feeding with exclusive breastfeeding exposing children to inadequate nutrient intake.

Child factors. Low birth weight (<2,500 g) remains a strong predictor of stunting, reflecting the lasting effects of intrauterine undernutrition and health insults [40,52]. These infants are more susceptible to respiratory issues, immunodeficiency, and metabolic disorders, which impair growth and development [53]. Gender disparities are also evident, with male children exhibiting higher stunting rates—a pattern that persists across various settings [34–36,39]. Biological differences, higher caloric needs, and greater susceptibility to preterm birth among males may explain this trend [54–57].

Child age is another significant factor; infants under six months tend to have lower stunting rates due to exclusive breastfeeding. However, stunting often becomes prominent after this period, particularly between 24–59 months, associated with inadequate complementary feeding, diarrheal diseases, and intestinal infections—factors that intensify during this developmental window [48,58–62]. Longitudinal data suggest that stunting peaks around 2–3 years and gradually declines afterward, reflecting the cumulative impact of nutrition and infections over time [63].

Household and Environmental Influences. Socioeconomic status is a well-established determinant; households with lower income levels face multiple nutritional and sanitary challenges that predispose children to stunting [64–68]. Even after adjusting for wealth, poor sanitation remains a crucial risk factor. Inadequate sanitation and hygiene practices facilitate infections that impair nutrient absorption and promote growth faltering [67]. Handwashing with soap and access to clean water significantly reduce infection-related growth delays.

Regional Variations and Sociocultural Factors. Regional disparities in stunting within Tanzania are evident and are often linked to socioeconomic and infrastructural factors. Highland areas and regions like Iringa and Njombe face unique challenges, including limited access to clean water, healthcare, and sanitation, compounded by socioeconomic constraints and cultural practices. For example, in Njombe, heavy workloads among women, limited male involvement, alcohol

consumption, and poor sanitation contribute markedly to high stunting rates [19–21,69,70]. Dietary monotony—favoring carbohydrate-rich staples like ugali and beans—exacerbates micronutrient deficiencies, further impairing growth.

## Limitations

The study faced several limitations. First and foremost, the cross-sectional nature of the data collection restricts the ability to establish causal relationships between variables. Since the first DHS, there have been changes in methodology, e.g. cut-off points for stunting. To address this, all data were analysed using the new WHO cut-off points for comparison. Additionally, there is a possibility that not all confounders were addressed since no new primary data were collected. A review of the DHS database revealed that the data collection tool does not include information on responsive care, and there are only a limited number of variables related to the safety and security domain. Furthermore, the data is insufficient regarding social care services and the support provided by family and foster care compared to institutional care.

## Conclusion and recommendations

Although child stunting has decreased in the country, it remains higher than the global average of 22.0%. The nation's stunting rates raise serious concerns about the development and health of children. Stunting is a multifaceted challenge that requires a multi-sectoral approach. It is essential to address region-specific challenges, gender disparities, economic status, and other maternal, child, and household factors. Integrating multisectoral interventions involves coordinating actions across health, agriculture, education, and social protection sectors to tackle factors contributing to stunting, such as poor water, sanitation, and hygiene, socioeconomic inequalities, and gender disparities. Furthermore, current regulations and standards place more emphasis on educating mothers about diet and health than on empowering them economically.

## Author contributions

**Conceptualization:** Ester Elisaria, Laura van der Erve, Bet Caeyers, Esther Nkuba, August Kuwawenaruwa.

**Formal analysis:** August Kuwawenaruwa.

**Funding acquisition:** Ester Elisaria, Laura van der Erve, Bet Caeyers.

**Investigation:** Ester Elisaria, Esther Nkuba.

**Methodology:** Esther Nkuba, August Kuwawenaruwa.

**Writing – original draft:** Ester Elisaria, Laura van der Erve, August Kuwawenaruwa.

**Writing – review & editing:** Ester Elisaria, Laura van der Erve, Bet Caeyers, Esther Nkuba.

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
