## [Decision Letter · Decision Letter 0]

Dear Dr. Kuwawenaruwa,

Thank you for submitting your manuscript to PLOS ONE. After careful consideration, we feel that it has merit but does not fully meet PLOS ONE’s publication criteria as it currently stands. Therefore, we invite you to submit a revised version of the manuscript that addresses the points raised during the review process.

**The manuscript will need a reorganization. First, authors present Findings and include references. This is not a common practice in Plos ONE. As a consequence, the Discussion is skeletal and needs flesh out. Second, the manuscript needs to clarify variables levels and all Tables to be presented as such. In its current form, it mixes all-level variables which is not easy to follow. Third, other comments in the annotated manuscript.**

We look forward to receiving your revised manuscript.

Kind regards,

Zacharie Tsala Dimbuene, Ph.D.

Academic Editor

PLOS ONE

“This project was funded with UK International Development from the UK government and New Zealand’s Ministry of Foreign Affairs and Trade (MFAT) contract PO number 10087; however, the views expressed do not necessarily reflect the UK or New Zealand’s government official policies”

3. In the online submission form, you indicated that [The dataset(s) supporting the conclusions of this article is owned by the Demographic and Health Surveys (DHS) Program, ICF, 530 Gaither Road, Suite 500, Rockville, MD 20850, USA and available upon request.].

Reviewers' comments:

Reviewer's Responses to Questions

**Comments to the Author**

1. Is the manuscript technically sound, and do the data support the conclusions?

Reviewer #1: Partly

Reviewer #2: No

2. Has the statistical analysis been performed appropriately and rigorously?

Reviewer #1: Yes

Reviewer #2: No

3. Have the authors made all data underlying the findings in their manuscript fully available?

Reviewer #1: Yes

Reviewer #2: Yes

4. Is the manuscript presented in an intelligible fashion and written in standard English?

Reviewer #1: Yes

Reviewer #2: No

Reviewer #1: I have uploaded a file with my comments. Overall, the study is well-conducted, but the methodology section lacks important details. The authors are advised to revise this section to include the missing information. Additionally, the entire manuscript should be reviewed and revised to address the identified issues.

Reviewer #2: The manuscript examines the changes in stunting among children under five in Tanzania in the last 30 years. While this trend analysis is important is understanding trend in stunting among children, the style of presenting the results makes the manuscript unnecessarily long and difficult to follow. The manuscript also requires a through edit for sentence structure in some paragraphs. Some detailed comments below.

Abstract

1. In the sentence “The frequency of stunted children was higher in households with the lowest socioeconomic status”, frequency should be percentage rather

2. “was significant associated” significant should be significantly

3. What is percentage change(decrease/increase) in stunting in Mainland Tanzania within the specified time? What is rate of change as well? Since the main objective was examine the change, the authors should consider specifying the rate of change

4. In the abstract, the author need to clarify that the analysis of the factors associated with stunting was done only for the 2022 data

Introduction

Well written

Methods

5. Data source: Did you obtain the actual DHS data for analysis, or you are simply reporting data/results presented in the Tanzania DHS report? The last sentence in this sub-heading “All data presented in the graphs as cross-sectional data points and tables are from the Tanzania DHS, 1991/92 – 2022” suggest that you are simply presenting results from the DHS reports. This needs clarity

6. This sentence is unnecessary in my opinion since the present manuscript has nothing to do with biomarkers or urine. “All biomarker questionnaires and urine specimens were dispatched weekly to the nearest TFNC laboratory.”

7. It is unclear from the statistical analysis if the authors considered strata and cluster effects in the analysis. The DHS data usually multistage sampling with clusters and strata and weighting is usually calculated with resulted reported as weighted. It will therefore be very appropriate if survey regression procedure accounting for the cluster and strata effects will be used for the analysis.

8. Consider re-writing for clarity “Both backward and forward elimination have been used to arrive at the final model for statistically significant le”

Results

9. “In Figure 1A, we can see that at least some of the policies and guidelines introduced over time have likely contributed to an improvement in the general trends in stunting in Tanzania;”. This sentence looks more like a discussion point rather than results. I suggest just describing what you observed from the analysis without making assumptions from the results here

10. Overall, the style of presenting the results is a mixed of results and discussion while a separate discussion heading remains making the manuscript unnecessarily long. Can the authors simply summary their results in 3-4 pages and then write 4–5-page discussion of their main findings. This will make the flow and reading easier to follow

Discussion

11. Another limitation to add is that since the data is cross-sectional, causality cannot be attributed to the associations presented for the predictors of stunting in 2022

**Do you want your identity to be public for this peer review?** For information about this choice, including consent withdrawal, please see our Privacy Policy

Reviewer #1: **Yes: ** Amon Exavery

Reviewer #2: No

---

## [Decision Letter · Decision Letter 1]

Dear Dr. Kuwawenaruwa,

We look forward to receiving your revised manuscript.

Kind regards,

Susan Horton

Academic Editor

PLOS ONE

Journal Requirements:

Additional Editor Comments:

I am incoming to replace the previous Academic Editor. My suggestions are:

1. Please fill out a STROBE checklist and supply to PLOS. (This does not need to be in the published article).

2. Please provide a written response to Reviewer 3: I was not able to find this in the online files, but I apologize in advance if it is there and I missed it.

3. Please follow Reviewer 3's request to shorten the discussion. Good practice is to state a point based on your results and succinctly compare to other articles in the literature.

4. Please also remove the list of abbreviations. This is not the style for this journal. You simply spell out the abbreviation on first use, and then after that just use the abbreviation.

5. I thought that PLOS required line numbers on a revised version. Please include these.

6. I have also provided a list of typos, as follows:

Abstract: suggest “maternal secondary education” rather than “secondary education”

P5: top line: consider saying “only” 19% of children aged…..”

Section entitled “TDHS sampling of households”: In first sentence, consider saying “took” rather than “considered”

Table 1: number of ANC visits: one of these categories should be either “greater than or equal to” or “less than or equal to” (later it seems that the first category is “less than or equal to 4”; the same issue is observed in table 2 and table 3.

P13, section on stunting and diarrhoea: it would be helpful to specify the period referred to in terms of episodes of diarrhoea, i.e. the question asked by DHS

P13: discussion of regions: given that the Abstract mentions Southwest Highlands and Southern Highlands specifically, it would be helpful to identify which regions belong in each of these Highland areas (Google didn’t seem to distinguish Southern Highlands and Southwestern, but indicated that Iringa, Mbeya and Ruvuma, all listed here, were part of the Southern Highlands)

Table 3: birth order and birth interval should be spelled out in full

P22: reference to Qadri et al – suggest specify the country to which this study refers.

P23: I couldn’t find Mtongwa et al (2022) in the reference list, and in any case this reference should be given a numbered reference, not (2022).

P23, second last line: “Ideally” should be deleted.

P24, last paragraph: delete first mention of “influence”. Should read “Child age was found to influence stunting”.

P25: don’t give date of publication for Karlsson et al, since that isn’t consistent with the style for this journal.

P25: first sentence of last paragraph: Suggest rephrasing as “The household factors include socio-economic status which was significantly associated with…”

P26, in first complete paragraph: capitalize Infant (in Infant and Young Child Feeding guidelines)

Section on Limitations: do you mean that “no new primary data were collected”?

Reviewers' comments:

Reviewer's Responses to Questions

**Comments to the Author**

Reviewer #1: (No Response)

Reviewer #3: All comments have been addressed

2. Is the manuscript technically sound, and do the data support the conclusions?

Reviewer #1: Yes

Reviewer #3: No

3. Has the statistical analysis been performed appropriately and rigorously?

Reviewer #1: No

Reviewer #3: No

4. Have the authors made all data underlying the findings in their manuscript fully available?

Reviewer #1: Yes

Reviewer #3: Yes

5. Is the manuscript presented in an intelligible fashion and written in standard English?

Reviewer #1: Yes

Reviewer #3: No

Reviewer #1: I appreciate the authors for making substantial revisions to the manuscript in response to my earlier comments. Their efforts to improve the clarity and presentation of the study are commendable.

1. However, my major concern is that the term "multilevel" has been replaced with "multivariate." If this change was merely in terminology, without a corresponding revision in the analysis, then the approach remains incorrect. In revising the analysis, the authors should first declare the data as 'survey' in design through the 'svyset' command in Stata (as mentioned, Stata was used). I think it takes this form... "svyset [pweight=v005], psu(v021) strata(v022)". Following this, each syntax for both descriptive and multivariate analyses should begin with the 'svy:' prefix to account for the survey design.

Currently, the analysis does not account for the inherent clustering of DHS data. Given the survey design, it is essential to either conduct a weighted analysis that accounts for stratification and primary sampling units (PSUs), or to use a multilevel approach to properly address the hierarchical structure of the data. Ignoring clustering can lead to incorrect standard errors and potentially biased estimates.

I strongly encourage the authors to either incorporate survey weights and adjust for clustering, or implement a multilevel model that aligns with the hierarchical nature of the data. Clarifying these methodological choices in the manuscript would greatly strengthen the rigor and validity of the findings.

2. Consider using 'multivariable' instead of 'multivariate' because multiple predictors or covariates are being associated with a single outcome - stunting.

3. Consider marking p-values of '0.000' as 'p<0.001'

Reviewer #3: Thank you for your valuable feedback and insightful comments. The manuscript needs to revised sections to enhance clarity, ensuring logical flow and coherence throughout the manuscript. The methodology to provide additional details on data analaysis. The discussion need to strengthened the discussion by incorporating relevant literature. The study adheres to ethical guidelines, and necessary approvals have been obtained since the study was based on secondary data analysis.

**Do you want your identity to be public for this peer review?** For information about this choice, including consent withdrawal, please see our Privacy Policy

Reviewer #1: **Yes: ** Amon Exavery, PhD

Reviewer #3: **Yes: ** Sampurna Kakchapati

---

## [Author Response · Author response to Decision Letter 2]

18 Jun 2025

Response to reviewers has been attached as a file.

---

## [Editor Report · Decision Letter 2]

Thirty Years of Declining Stunting in Tanzania: Trends and Ongoing Challenges

PONE-D-24-53994R2

Dear Dr. Kuwawenaruwa,

We’re pleased to inform you that your manuscript has been judged scientifically suitable for publication and will be formally accepted for publication once it meets all outstanding technical requirements.

Kind regards,

Susan Horton

Academic Editor

PLOS ONE

Additional Editor Comments (optional):

Thank you for your careful response to the reviewers' comments, and for providing the STROBE checklist.
---

## [Editor Report · Acceptance letter]

PONE-D-24-53994R2

PLOS ONE

Dear Dr. Kuwawenaruwa,

I'm pleased to inform you that your manuscript has been deemed suitable for publication in PLOS ONE. Congratulations! Your manuscript is now being handed over to our production team.

Kind regards,

on behalf of

Dr. Susan Horton

Academic Editor

PLOS ONE